# Reproductive Ecology of Distylous Shoreside *Polygonum criopolitanum* Hance

**Ming-Lin Chen \*, Meng-Ying Qi, Bei-Bei Bai and Xue Han**

Provincial Key Laboratory of Conservation and Utilization of Biological Resources in Anhui,
Anhui Normal University, Wuhu 241000, China; qmengying2022@126.com (M.-Y.Q.); bei12457@163.com (B.-B.B.);
hanxue200403@163.com (X.H.)
\* Correspondence: chenminglin@ahnu.edu.cn

**Abstract:** In this study, distyly was clearly confirmed in *Polygonum criopolitanum* Hance, which exhibited strict self-incompatibility. Unlike other distylous species, style-morph ratios of *P. criopolitanum* often deviated obviously from 1:1, and many populations were solely composed of long or short stylous flowers; the 1:1 style-morph ratio was occasionally found in very large populations. *P. criopolitanum* was dimorphic for intrinsic features such as style height and anther height and ancillary features such as pollen size and number. The L-morph flowers produced a significantly smaller and higher number of pollen grains than the S-morph flowers, and the stigma papillae of both morphs were not significantly different. We nearly found no seed sets in most wild populations and very low seed sets occasionally occurred in large populations, which was different from other species of Polygonaceae. Mating experiments showed that *P. criopolitanum* has a strict self-incompatibility system and clonal propagation was more common than sexual propagation, which was adaptive with the unisexual wild populations. Hygrocolous habitat, 20–60% soil water content, and height gap less than 4 m to the adjacent water were the main limiting factors for the distribution of *P. criopolitanum*.

**Keywords:** *Polygonum criopolitanum*; distyly; style-morph ratios; seed sets; strict self-incompatibility

## 1. Introduction

Heterostyly is a genetic polymorphism in which plant populations are composed of 2 (distyly) or 3 (tristyly) floral morphs. The morphs exhibit reciprocal positioning of anthers and stigmas. Flowers with the L-styled morphology have a stigma(s) positioned above the anthers, whereas flowers with the S-styled morphology have anthers placed above the stigma(s) [1,2]. Besides the differences in floral morphology, distyly is often linked to a sporophytically controlled, diallelic incompatibility system that results in intramorph incompatibility [3].

Heterostyly has been documented in at least 193 genera of 30 angiosperm families [2,4]. Some families-genera possess hundreds of heterostylous species, e.g., Oxalidaceae-*Oxalis*, Primulaceae-*Primula*, and Rubiaceae [3], and the occurrence of heterostyly has been recently reported in *Perovskia* [5]. Distyly in Polygonaceae was first described by Hildebrand in *Fagopyrum esculentum* over 100 years ago [6]; subsequently, many distylous species have been reported for the genera *Oxygonum* and *Aconogonum* [7–10]. The type genus for the family, *Polygonum* (which has more than 300 species) [11], has a worldwide distribution, but it has been seldom reported to be heterostylous. The first distylous *Polygonum chinense* was documented in detail by Reddy et al. (1977) [8]. To date, some other distylous species in *Polygonum* have been reported [12–17].

*Polygonum criopolitanum* Hance is an annual herb with a tufted, prostrate stem at the base that grows to 10–15 cm. Its inflorescence is terminal and capitate, and the perianth is composed of 5-parted purplish-red tepals. The species is always distributed in the sand by riversides and wet ditches. Most of the previous studies on this species focused mainly on its ecology [18], mating system [19], and plant resources [20], and, to the best of

our knowledge, the reproductive characters of *P. criopolitanum* have never been reported in detail.

In this study, we examined variations in floral and reproductive traits and incompatibility systems in natural populations of *P. criopolitanum*. The aims of this study were as follows: (1) to determine whether *P. criopolitanum* is typically distylous as other documented distylous species; (2) to record the morph ratios of wild populations of *P. criopolitanum*; (3) to test the compatibility system after pollen germination, stigma receptivity, and seed set tests; and (4) to discuss the relationship between ecological factors and environmental factors by using canonical correlation analysis (CCA).

## 2. Materials and Methods

### 2.1. Study Materials

This study was conducted along the Yangzi River and in Anhui, Jiangxi Province (Figure 1). Herbarium specimens or living materials of *P. criopolitanum* from Anhui and Jiangxi Province, China, were used for this study. All vouchers were deposited at the herbarium of Anhui Normal University (ANUB), China.

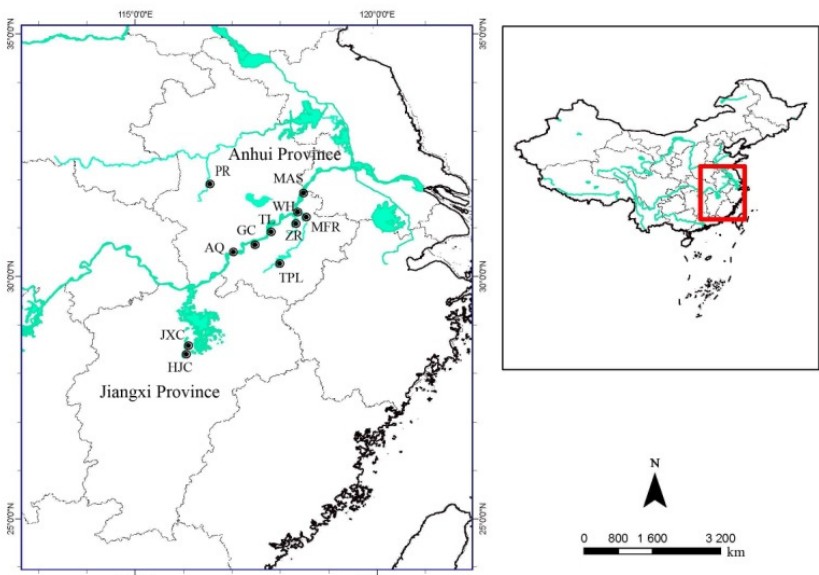

**Figure 1.** Study sites of *Polygonum criopolitanum* (●: Population. JXC = Jinxian County; HJC = Huangjiacun; TPL = Taiping Lake; MFR = Mafeng River; ZR = Zhang River; AQ = Anqing; GC = GuiChi; TL = Tongling; WH = Wuhu; MAS = Maanshan; PR =Pi River).

### 2.2. Study Methods

#### 2.2.1. Floral Characters

To document distyly in *P. criopolitanum*, the height of stigma, height of anther, stigma-anther separation, length of tepal, and flower diameter (sketched in Figure 2) were recorded for 50 or 100 flowers of each morph at random from the Wuhu Mafeng River population (Figure 1), using a vernier caliper with a resolution of 0.01 mm.

To count the number of pollen produced, anthers from 100 flowers per morph of the Wuhu populations were placed on individual microscope slides. The numbers of pollen grains per flower were counted using a light microscope, and the diameter of 10 moderate pollen grains from each flower was measured [12,21]. Pollen counts for the style morphs were compared using Student's *t*-test.

#### 2.2.2. Scanning Electron Microscopy

According to the standard acetolysis method [22], pollen grains were mounted in glycerin jelly and sealed with paraffin. The size of well-formed pollen grains from each sample was measured. Scanning electron microscopy was performed using acetolyzed

pollen grains coated with Au/Pd under a Hitachi S4800 scanning electron microscope (SEM). Fresh stigmas were mounted in a slow-drying glue and observed under the SEM.

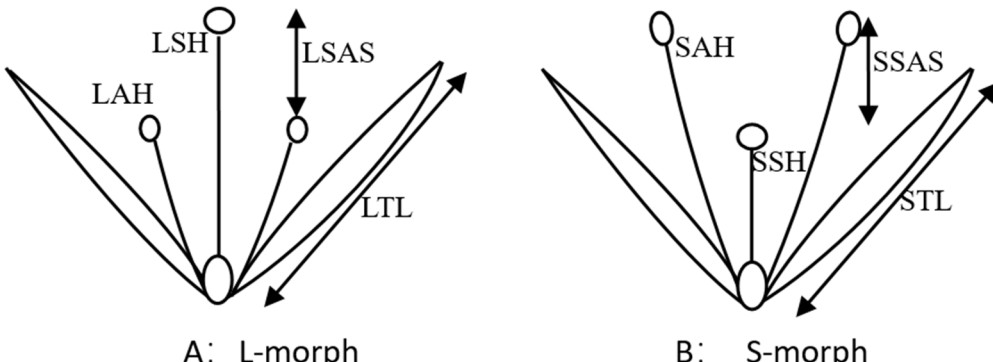

**Figure 2.** Pattern diagram of distylous flower of *P. criopolitanum* ((**A**): L-morph; (**B**): S-morph). LTL: tepal length of L-morph; LAH: anther height of L-morph; LSH: stigma height of L-morph; LSAS: stigma-anther separation of L-morph; STL: tepal length of S-morph; SAH: anther height of S-morph; SSH: stigma height of S-morph; SSAS: stigma-anther separation of S-morph.

### 2.2.3. Population Structure Survey

To determine the relative frequencies of the 2 style morphs in populations of *P. criopolitanum*, we surveyed 44 samples to test whether L- and S-morphs occurred at equal frequencies. The chi-square test was performed in R software [23]. We sampled all plants in sporadic populations and $1 \times 1$ m in patches of populations; for larger samples, and we performed $10 \times 10$ m sampling.

### 2.2.4. Environmental and Ecological Factors

Nineteen samples were obtained in Huangshan Taiping Lake, AnqingYingjiang Tower, Guichi Shibasuo, Tongling Shizishan, Yangzi River in Wuhu, Longwo Lake, and Zhang River. Four samples were obtained along the Yangzi River, Longwo Lake, and Zhang River. The first sample was located at the water side, and the other three samples were located along a line drawn perpendicular to the shore at 1-m intervals in altitude up the slope of the riverbank. The number of plants, height, relative coverage, species richness, and presence or absence of *P. criopolitanum* was recorded. The altitude, height gap to the adjacent water, habitat, annual precipitation, annual average temperature, slope, and canopy density of each sample were also documented, and1 kg of soil from each sample was transferred to the laboratory to test ecological factors such as soil water content, pH, organic matter, total nitrogen (TN), and total phosphorus (TP). The soil water content was measured using the GB9834-88 method; soil TN, Kjeldahl method; and soil TP, Mo-Sb colorimetric method [24].

### 2.2.5. CCA

CCA was performed using Canoco for Windows 4.5. The graph settings were confirmed using the data for the environmental and ecological factors.

### 2.2.6. Fluorescent Microscopy

To identify the extent of incompatibility of the breeding system of *P. criopolitanum*, pollen germination, and pollen tube growth were determined. The style morphs from each population were pollinated; we waited for 12 h to observe growth in the pollen tube, then they were stored in FAA until staining. To estimate the pollen germination, theharvested styles were softened in 8 mol/L NaOH for 24 h rinsed with distilled water. Then, they were stained for 4 h in 0.4 mg/mL aniline blue solution in phosphate buffer (pH 8.0) and gently squashed in a small drop of glycerol mounting medium under a cover slip. The pistil and pollen were scanned with an Olympus (BX61) epifluorescence microscope (420–470 nm

excitation, 490–535 nm emission). For each style, pollen germination was detected by the presence of a pollen tube projecting from the grain, and 10 replicates were performed.

### 2.2.7. Seed Set Experiments

The seed set experiments were performed using natural populations and controlled pollination populations. For the natural populations, we set 14 1 × 1 m samples in Taiping Lake (2 samples), Luan Pi River (1), Guichi (1), Anqing (1), Tongling (1), Wuhu (5), and Jinxian (3). We calculated the style-morph ratio and then obtained all infructescences of one plant to count the number of seeds ($n$ = 45).

The hybrid experiments were conducted in the laboratory. Legitimate and illegitimate pollination (including selfing) were performed; the seed sets were recorded in every treatment mode: emasculation and bagging, selfing, illegitimate (intramorph) pollination, and legitimate (intermorph) pollination. Each mode was conducted using 100 flowers.

### 2.2.8. Seed Germination Experiments

Normally developed seeds were soaked in water for 48 h and then disinfected in 75% ethyl alcohol for 30 s. Thirty treated seeds were placed in a culture dish in an illumination incubator to observe the seed germination, with 3 replicates.

## 3. Results

### 3.1. Floral Biology

*P. criopolitanum* is an annual herb with terminal and capitate inflorescence. The peduncle is covered by dense glandular hair, and each bract is 1-flowered. The pattern of floral variation in the wild populations demonstrates that *P. criopolitanum* has conventional distylous floral syndrome (Figure 3A–D). The tepal lengths of the long stylous flower (hereafter L-morph) and short stylous flower (hereafter S-morph) were 2.53 ± 0.44 and 2.47 ± 0.29 mm, respectively, and the tepal diameters of the L-morph and S-morph were 6.58 ± 0.47 and 6.42 ± 0.25 mm, respectively. No significant differences were observed between the tepal lengths as well tepal diameters of the L-morph and S-morph (Table 1, $p$ > 0.05). The anthers and stigma are reciprocally positioned in the flowers of the two morphs (Figures 4 and 5). Five stamens are situated between the base of adjacent tepals, and the anthers are purple. Styles 2, seldom 3, connate at the middle-upper part (Figure 5A–D). In addition, the two morphs have five nectaries arranged at the base of each ovary (Figure 3C,D). The stigma and anther heights of the L-morph were 4.02 ± 0.66 and 2.19 ± 0.42 mm, respectively, and the stigma and anther heights of the S-morph were 3.90 ± 0.42 and 1.83 ± 0.55 mm, respectively; significant differences were observed between the heights of the stigma and anther of both morphs ($p$ < 0.05, Table 1). We found that the stigma-anther separation of the L-morph (2.06 ± 0.39) was longer than that of the S-morph (1.83 ± 0.55; $p$ < 0.05, Table 1).

The stigma, which is capitate or spherical with globular papillae on the stigma surface, is similar in the L-morph and S-morph. No significant differences were observed in the size and shape of the stigma papillae (Figure 5C–F).

The pollen grains of *P. criopolitanum* are spheroidal in both morphs, and the pollen grain surface is covered with reticulate exine structures that are pentagonal or hexagonal. Each reticular mesh in the pollen contains many smooth papillae (Figure 5G–J).

The pollen size and number of the two morphs of *P. criopolitanum* showed significant differences. The mean pollen diameters of the L-morph and S-morph were 51 ± 1.92 μm and 62 ± 2.51 μm, respectively, and the mean pollen number per flower of the L-morph and S-morph was 647 ± 40 and 526 ± 38, respectively (Table 1). Although an overlap was detected in the pollen sizes of both morphs, flowers of the L-morph produced significantly more and smaller pollen grains than the ones of the S-morph (Table 1, $p$ < 0.01).

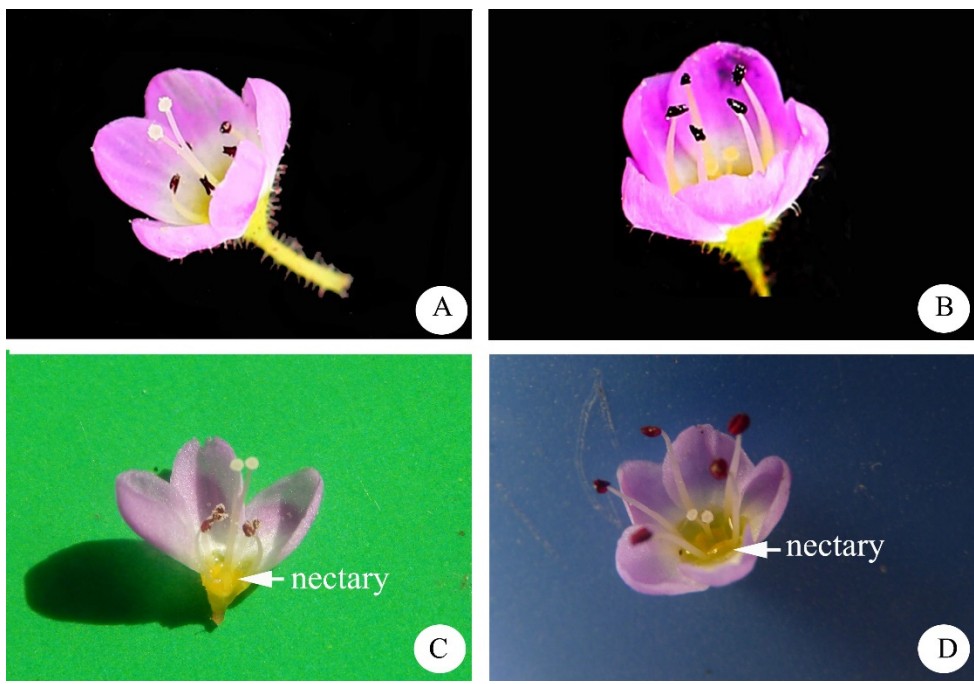

**Figure 3.** Flowers of the L- and S-morphs of *P. criopolitanum* ((**A**,**C**): L-morph; (**B**,**D**): S-morph).

**Table 1.** Morphological features of the long- and short-morph flowers of *P.criopolitanum*. Differences between the means were analyzed using one-way analysis of variance (ANOVA; mean ± standard error).

| Flower Characteristic | L-Morph | S-Morph | *p*-Value | Sample Number |
|---|---|---|---|---|
| Length of tepal(mm) | 2.52 ± 0.44 | 2.47 ± 0.29 | >0.05 | 50 |
| Flower diameter(mm) | 6.58 ± 0.47 | 6.42 ± 0.25 | >0.05 | 50 |
| Height of stigma(mm) | 4.02 ± 0.66 | 1.84 ± 0.26 | <0.001 | 50 |
| Height of anther(mm) | 2.19 ± 0.42 | 3.90 ± 0.42 | <0.001 | 50 |
| Stigma-anther separation(mm) | 2.06 ± 0.39 | 1.83 ± 0.55 | <0.05 | 100 |
| Pollen number | 647 ± 40 | 526 ± 38 | <0.01 | 100 |
| Pollen diameter(μm) | 51 ± 1.92 | 62 ± 2.51 | <0.001 | 100 |

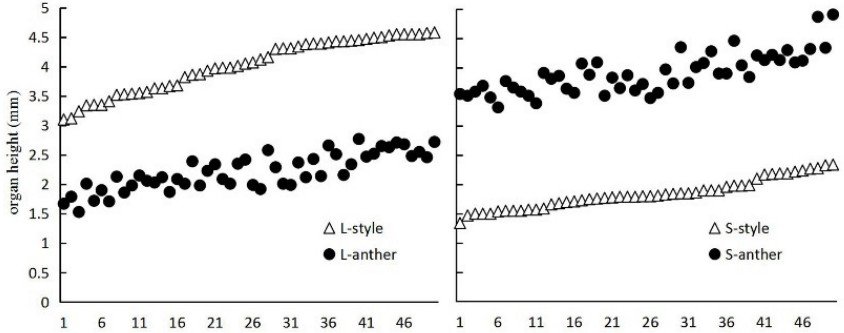

**Figure 4.** Style and anther length of *P.criopolitanum* (ranked by style length to illustrate the reciprocal correspondence of stigma and anther positions in the long- and short-styled morphs. Positions of the stigmas and anthers are indicated by open triangle (△) and solid circle (●), respectively.

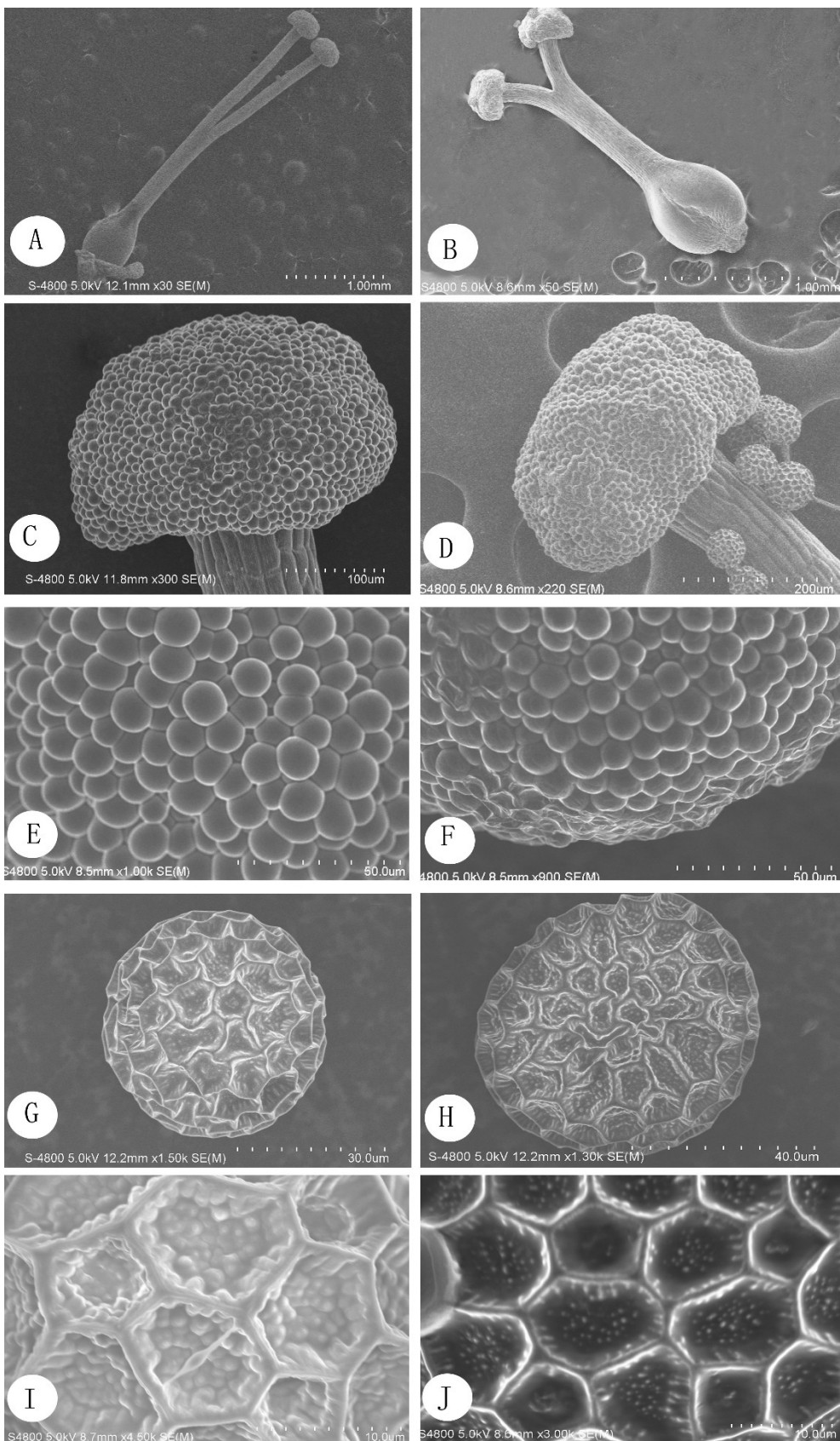

**Figure 5.** Micromorphological characteristics of *P. criopolitanum* (SEM). (**A**) pistil of L-morph; (**B**) pistil of S-morph; (**C**) stigma of L-morph; (**D**) stigma of S-morph; (**E**) stigma papillae of L-morph; (**F**) stigma papillae of S-morph; (**G**) pollen of L-morph; (**H**) pollen of S-morph; (**I**) pollen epidermal ornamentation of L-morph; (**J**) pollen epidermal ornamentation of S-morph.

### 3.2. Style-Morph Ratios

The chi-square test is a measure of whether the style-morph ratios deviate from 1:1. The survey results showed that the style-morph ratios of *P. criopolitanum* obviously deviated from 1:1, and parthenogenetic populations were always detected in the wild populations. We found that many populations were solely composed of long or short stylous flowers (e.g., ZhengfengTower or Jinxian County populations), whereas the 1:1 morph ratio was seldom found in larger populations (Jinxian County, Taiping Lake, and Mafeng River populations; Table 2).

**Table 2.** Style-morph ratios in 44 natural populations of *P. criopolitanum*.

| Geographic Position | Sample Sites | Sample Size (m²) | Flower No. of L-Morph | Flower No. of S-Morph | L-Morph:S-Morph | $\chi^2$ | *p*-Value | Deviate from 1:1 |
|---|---|---|---|---|---|---|---|---|
| Anqing City | ZhengfengTower | 1 | 211 | 0 | - | 211 | $2.2 \times 10^{-16}$ | Y |
| | Zongyang County | 1 | 185 | 0 | - | 185 | $2.2 \times 10^{-16}$ | Y |
| Guichi City | Huamiao | 1 | 12 | 0 | - | 12 | $5.32 \times 10^{-4}$ | Y |
| | Shibasuo | 1 | 0 | 97 | - | 97 | $2.2 \times 10^{-16}$ | Y |
| Tongling City | Shizishan 1 | 1 | 0 | 37 | - | 37 | $1.18 \times 10^{-9}$ | Y |
| | Shizishan 2 | 1 | 0 | 285 | - | 285 | $2.2 \times 10^{-16}$ | Y |
| Wuhu City | MafengRiver | 100 | 785 | 721 | 1.08 | 2.72 | $9.91 \times 10^{-2}$ | N |
| | MafengRiver 1 | 1 | 0 | 63 | - | 63 | $2.07 \times 10^{-15}$ | Y |
| | MafengRiver 2 | 1 | 82 | 0 | - | 82 | $2.2 \times 10^{-16}$ | Y |
| | MafengRiver 3 | 1 | 46 | 0 | - | 46 | $1.18 \times 10^{-11}$ | Y |
| | QingyiRiver | 1 | 0 | 81 | - | 81 | $2.2 \times 10^{-16}$ | Y |
| | Zhang River 1 | 100 | 102 | 0 | - | 102 | $2.2 \times 10^{-16}$ | Y |
| | Zhang River 2 | 1 | 150 | 162 | 0.93 | 0.46 | 0.50 | Y |
| | ZhaojiaRiver | 1 | 72 | 0 | - | 72 | $2.2 \times 10^{-16}$ | Y |
| | LongwoLake | 1 | 0 | 95 | - | 95 | $2.2 \times 10^{-16}$ | Y |
| | Wanzhi | 1 | 0 | 89 | - | 89 | $2.2 \times 10^{-16}$ | Y |
| Manshan City | Yushanqu 1 | 1 | 19 | 0 | 1 | 19 | $1.31 \times 10^{-5}$ | Y |
| | Yushanqu 2 | 1 | 36 | 0 | 1 | 36 | $1.97 \times 10^{-9}$ | Y |
| | Yushanqu 3 | 1 | 22 | 16 | 1.38 | 0.95 | 0.33 | N |
| | Yushanqu 4 | 1 | 21 | 0 | 1 | 21 | $4.59 \times 10^{-6}$ | Y |
| | Yushanqu 5 | 1 | 18 | 92 | 0.20 | 49.78 | $1.72 \times 10^{-12}$ | Y |
| Tunxi City | Taiping Lake | 100 | 340 | 316 | 1.08 | 0.88 | 0.35 | N |
| | Taiping Lake 1 | 100 | 54 | 0 | - | 54 | $2 \times 10^{-13}$ | Y |
| | Taiping Lake 2 | 1 | 31 | 0 | - | 31 | $2.58 \times 10^{-8}$ | Y |
| | Taiping Lake 3 | 1 | 0 | 49 | - | 49 | $2.56 \times 10^{-12}$ | Y |
| | Taiping Lake 4 | 1 | 11 | 0 | - | 11 | $9.11 \times 10^{-4}$ | Y |
| Luan City | Luan Pi River | 100 | 345 | 67 | 5.15 | 187.58 | $2.2 \times 10^{-16}$ | Y |
| | Luan Pi River1 | 1 | 121 | 0 | - | 121 | $2.2 \times 10^{-16}$ | Y |
| | Luan Pi River2 | 1 | 56 | 0 | - | 56 | $2.2 \times 10^{-16}$ | Y |
| | Luan Pi River3 | 1 | 0 | 160 | - | 160 | $2.2 \times 10^{-16}$ | Y |
| | Luan 1 | 1 | 42 | 24 | 1.75 | 4.91 | 0.03 | Y |
| | Luan 2 | 1 | 35 | 0 | 1 | 35 | $3.30 \times 10^{-9}$ | Y |
| | Luan 3 | 1 | 33 | 18 | 1.83 | 4.41 | 0.04 | Y |
| Nanchang City | Jinxian County | 100 | 895 | 842 | 1.06 | 1.62 | 0.20 | N |
| | Jinxian County 1 | 100 | 112 | 0 | - | 112 | $2.2 \times 10^{-16}$ | Y |
| | Jinxian County 2 | 1 | 56 | 12 | 4.67 | 28.47 | $9.51 \times 10^{-9}$ | Y |
| | Jinxian County 3 | 1 | 0 | 90 | - | 90 | $2.2 \times 10^{-16}$ | Y |
| | Huangjiacun 1 | 1 | 34 | 11 | 3.09 | 11.76 | $6.1 \times 10^{-4}$ | Y |
| | Huangjiacun 2 | 1 | 0 | 34 | 1 | 34 | $2.2 \times 10^{-16}$ | Y |
| | Huangjiacun 3 | 1 | 26 | 0 | 1 | 26 | $3.41e^{-7}$ | Y |
| | Huangjiacun 4 | 1 | 43 | 0 | 1 | 43 | $5.47 \times 10^{-11}$ | Y |
| | Huangjiacun 5 | 1 | 27 | 30 | 0.90 | 0.16 | 0.69 | N |
| | Huangjiacun 6 | 1 | 0 | 30 | 1 | 30 | $4.32 \times 10^{-8}$ | Y |
| | Huangjiacun 7 | 1 | 29 | 27 | 1.07 | 0.07 | 0.79 | N |

### 3.3. CCA Results

3.3.1. Characteristics of the Environmental Factors

The environmental factor indices of 19 samples are listed in Table 3.

**Table 3.** Characteristics of the environmental factors of *P. criopolitanum*.

| Sample | Population No. | Altitude (m) | Height Gap to the Adjacent Water (m) | Soil Water Content | Habitat (1: Aquatic, 2: Hygrocolous, 3: Xeromorphic) | Annual Precipitation (mm) | Annual Average Temperature (°C) | Slope (°C) | Canopy Density | pH | Organic Matter (g/kg) | Total Nitrogen (g/kg) | Total Phosphorus (g/kg) |
|---|---|---|---|---|---|---|---|---|---|---|---|---|---|
| | | alt | heig | wat | hab | pre | tem | slo | cad | pH | orm | ton | top |
| WushiTown | 1 | 18 | 0.5 | 0.39 | 2 | 1564.5 | 15.4 | 6 | 0.7 | 6.56 | 38.5 | 0.409 | 0.208 |
| Taiping Lake | 2 | 17 | 0.8 | 0.38 | 2 | 1564.5 | 15.4 | 12 | 0.56 | 6 | 50.4 | 0.254 | 0.45 |
| AnqingYingjiangTower | 3 | 21 | 1 | 0.32 | 2 | 1400 | 15.5 | 49 | 0.1 | 5.6 | 48.6 | 0.102 | 0.345 |
| Guichishibasuo | 4 | 17 | 1.2 | 0.25 | 2 | 1578.3 | 15.9 | 8 | 0.32 | 5.72 | 19.5 | 0.393 | 0.41 |
| Tonglingshizishan | 5 | 156 | 2.1 | 0.34 | 2 | 1589.1 | 15.8 | 32 | 0.38 | 7 | 22.7 | 0.327 | 0.208 |
| Yangzi River 1 | 6 | 6.5 | 0 | 1 | 1 | 1054.8 | 16.2 | 12 | 0 | 7.32 | 33.5 | 0.256 | 0.467 |
| Yangzi River 2 | 7 | 7.5 | 1 | 0.38 | 2 | 1054.8 | 16.2 | 25 | 0.75 | 6.33 | 21.9 | 0.278 | 0.285 |
| Yangzi River 3 | 8 | 9.5 | 3 | 0.32 | 2 | 1054.8 | 16.2 | 25 | 0.76 | 6.38 | 34.6 | 0.346 | 0.156 |
| Yangzi River 4 | 9 | 11.5 | 5 | 0.09 | 3 | 1054.8 | 16.2 | 25 | 0.56 | 6.36 | 34.9 | 0.165 | 0.234 |
| Zhang River 1 | 10 | 9 | 0 | 1 | 1 | 1054.8 | 16.2 | 6 | 0 | 7.39 | 44.6 | 0.287 | 0.35 |
| Zhang River 2 | 11 | 11 | 2 | 0.519 | 2 | 1054.8 | 16.2 | 23 | 0.63 | 7.17 | 29.3 | 0.158 | 0.4 |
| Zhang River 3 | 12 | 12 | 3 | 0.485 | 2 | 1054.8 | 16.2 | 25 | 0.42 | 7.14 | 28.6 | 0.402 | 0.321 |
| Zhang River 4 | 13 | 13 | 4 | 0.123 | 3 | 1054.8 | 16.2 | 26 | 0.35 | 7.52 | 44.7 | 0.483 | 0.235 |
| LongwoLake 1 | 14 | 10 | 0 | 1 | 1 | 1054.8 | 16.2 | 42 | 0 | 7.23 | 45.6 | 0.306 | 0.408 |
| LongwoLake 2 | 15 | 11 | 1 | 0.34 | 2 | 1054.8 | 16.2 | 12 | 0.23 | 7.26 | 48.9 | 0.415 | 0.296 |
| LongwoLake 3 | 16 | 13 | 3 | 0.26 | 2 | 1054.8 | 16.2 | 28 | 0.25 | 7.42 | 21.3 | 0.439 | 0.318 |
| LongwoLake 4 | 17 | 15 | 5 | 0.45 | 3 | 1054.8 | 16.2 | 12 | 0.27 | 7.05 | 20.5 | 0.308 | 0.375 |
| Pi River 1 | 18 | 56 | 1.5 | 0.29 | 2 | 1100 | 16.3 | 21 | 0.32 | 5.67 | 42.6 | 0.446 | 0.284 |
| Pi River 2 | 19 | 62 | 5 | 0.03 | 3 | 1100 | 16.3 | 19 | 0.29 | 5.62 | 38.6 | 0.479 | 0.368 |

### 3.3.2. Characteristics of Ecological Factors

The ecological factor indices of 19 samples are listed in Table 4.

**Table 4.** Characteristics of ecological factors of *P. criopolitanum*.

| Samples | Sample Number | Plants Number | Relative Coverage | Height(cm) | Relative Frequency | Species Richness | Exist or Not |
|---|---|---|---|---|---|---|---|
| Wushi Town | 1 | 118 | 0.85 | 16 | 0.76 | 3 | 1 |
| Taiping Lake | 2 | 12 | 0.34 | 15 | 0.54 | 4 | 1 |
| AnqingYingjiangTower | 3 | 4 | 0.16 | 14 | 0.12 | 1 | 1 |
| GuichiShibasuo | 4 | 3 | 0.12 | 11 | 0.21 | 2 | 1 |
| TonglingShizishan | 5 | 4 | 0.15 | 13 | 0.18 | 5 | 1 |
| Yangzi River 1 | 6 | 0 | 0 | 0 | 0 | 0 | 0 |
| Yangzi River 2 | 7 | 28 | 0.8 | 15 | 0.76 | 2 | 1 |
| Yangzi River 3 | 8 | 39 | 0.82 | 14 | 0.82 | 5 | 1 |
| Yangzi River 4 | 9 | 0 | 0 | 0 | 0 | 8 | 0 |
| Zhang River 1 | 10 | 0 | 0 | 0 | 0 | 1 | 0 |
| Zhang River 2 | 11 | 105 | 0.82 | 17 | 0.76 | 4 | 1 |
| Zhang River 3 | 12 | 90 | 78 | 16 | 0.74 | 8 | 1 |
| Zhang River 4 | 13 | 0 | 0 | 0 | 0 | 1 | 0 |
| Longwo Lake 1 | 14 | 0 | 0 | 0 | 0 | 0 | 0 |
| Longwo Lake 2 | 15 | 7 | 0.13 | 15 | 0.26 | 5 | 1 |
| Longwo Lake 3 | 16 | 4 | 0.08 | 15 | 0.21 | 4 | 1 |
| Longwo Lake 4 | 17 | 0 | 0 | 0 | 0 | 4 | 0 |
| Pi River 1 | 18 | 29 | 0.76 | 16 | 0.35 | 6 | 1 |
| Pi River 2 | 19 | 0 | 0 | 0 | 0 | 12 | 0 |

Note: pln: plants number; rec: relative coverage; heig: height; ref: relative frequency; spr: species richness; exn: exist or not; 1 = exist, 0 = absence.

### 3.3.3. CCA of *P. criopolitanum* Samples

The relationship between ecological factors and environmental factors is shown in Figure 6.

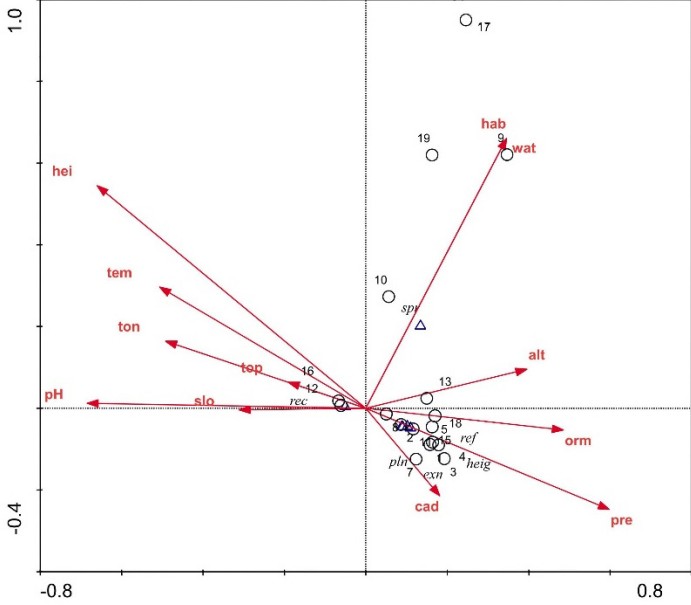

**Figure 6.** CCA ordination diagram of *P. criopolitanum*. Note: alt:altitude; hei: height gap; wat: water; hab: habitat; pre: precipitation; tem: temperature; slo: slope; cad: canopy density; pH: pH; orm: organic matter; ton: total nitrogen; top: total phosphorus. Positions of the samples and environmental factors are indicated by open circle (○) and open triangle (△), respectively.

In the CCA ordination diagram, the red lines and arrowheads refer to the environmental factors, and the length of the segment refers to the relationship degree between the sample distribution and environmental factors. The angle between the ordination axes and arrowhead connecting the line indicates the correlation degree between the environmental factor and its ordination axes, and the quadrant where the arrowhead is distributed indicates the positive or negative relationship between the environmental factor and its ordination axes. Figure 6 shows the major ecological factors that affect the population distribution of *P. criopolitanum*: habitat, soil water content, and height gap to the adjacent water. Hygrocolous habitat, 20–60% soil water content, and height gap less than 4 m to the adjacent water were the main limiting factors for the distribution of *P. criopolitanum*. Annual precipitation maydirectly influence the soil water content and hygrocoloushabitat. The population distribution of *P. criopolitanum* had no obvious relationship with the slope, pH, altitude, organic matter, TN, and TP. The axes showed that the soil water content increased progressively from top to bottom.

### 3.4. Mating System Relationships

The fluorescent experiments showed that the rate of pollen germination in the style of the intermorph was obviously higher than that of the self- and intramorph (Figure 7A–F). The statistics showed that the pollen germination rate of pin style × thrum pollen was $72.0 \pm 4.0\%$ ($n = 10$) and that of thrum style × pin pollen was $73.0 \pm 3.5\%$ ($n = 10$). During intramorph pollination, the pollen germination rate of pin style × pin pollen was $43 \pm 2.6\%$ ($n = 10$) and that of thrum style × thrum pollen was $42.0 \pm 3.1\%$ ($n = 10$).

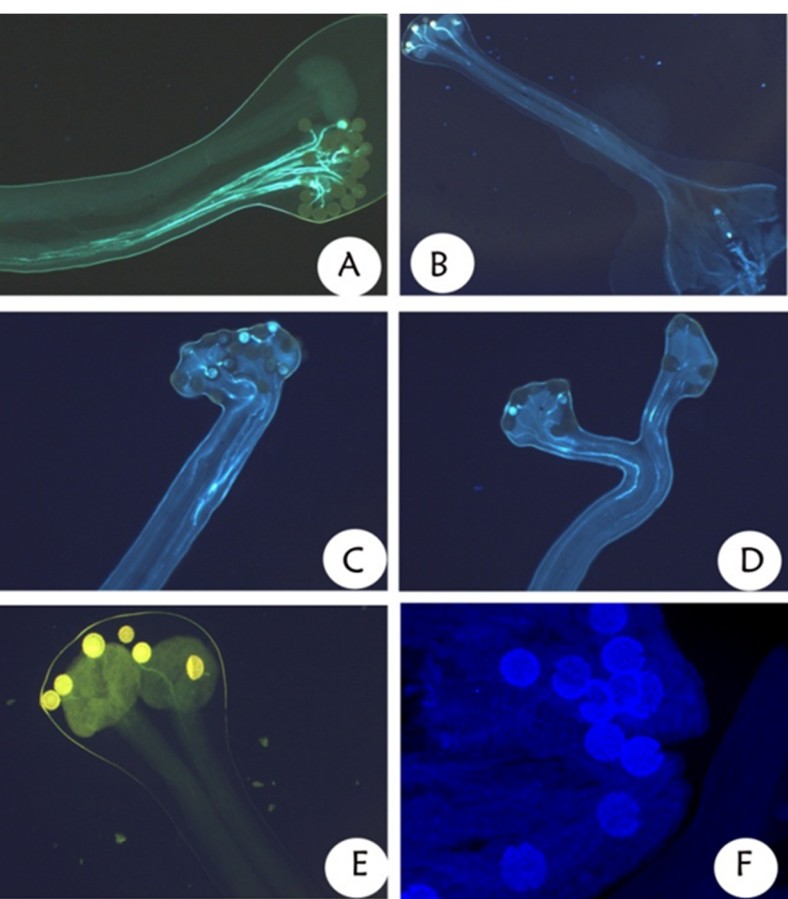

**Figure 7.** Fluorescent experiments of pollen germination of *P. criopolitanum*. (**A**) L-morph intermorph pollinations; (**B**) S-morph legitimate pollinations; (**C**) L-morph illegitimate pollination; (**D**) S-morph illegitimate pollination. (**E**) Selfing of L-morph; (**F**) Selfing of S-morph.

During self-pollination, the rate of pollen germination of the pin self was $9.0 \pm 0.7\%$, whereas no pollen germination was identified in the case of the thrum self.

*3.5. Seed Sets*

The statistics showed that the seed sets of *P. criopolitanum* were very low in the wild populations; moreover, the seed sets in the population were often 0, and, even in larger populations with both L- and S-morph, the seed sets were always less than 10% (Table 5).

**Table 5.** Seed sets of *P.criopolitanum* in different populations under natural conditions.

| Population | Population | | Fruit Sets |
|---|---|---|---|
| | **L-Morph** | **S-Morph** | |
| Taipinghu 1 | 156 | 32 | 4.17 |
| Taipinghu2 | 58 | 0 | 0 |
| Luan Pi River | 145 | 0 | 0 |
| Guichi Shibasuo | 0 | 45 | 0 |
| Anqing Zhengfeng Tower | 32 | 0 | 0 |
| Tongling | 49 | 0 | 0 |
| Wuhu MafengRiver | 176 | 152 | 3.16 |
| Wuhu Qingyijiang | 65 | 0 | 0 |
| Wuhu Longwo Lake | 0 | 47 | 0 |
| Wuhu Zhang River 1 | 89 | 0 | 0 |
| Wuhu Zhang River 2 | 80 | 11 | 0 |
| Jinxian County 1 | 215 | 208 | 6.67 |
| Jinxian County 2 | 0 | 73 | 0 |
| Jinxian County 3 | 87 | 0 | 0 |

The statistics showed that the seed set rate was 0 under emasculation, selfing, and illegitimate pollination conditions. Moreover, the seed set rate was low under legitimate pollination (Table 6).

**Table 6.** Seed set rate under different treatment conditions.

| Process Mode | Pin | | | Thrum | | |
|---|---|---|---|---|---|---|
| | **Flowers** | **Seeds** | **Seed Sets (%)** | **Flowers** | **Seeds** | **Seed Sets (%)** |
| Emasculation, bagged | 100 | 0 | 0 | 100 | 0 | 0 |
| Selfing | 100 | 0 | 0 | 100 | 0 | 0 |
| Illegitimate pollination (intramorph) | 100 | 0 | 0 | 100 | 0 | 0 |
| Legitimate pollination | 100 | 3 | 3% | 100 | 5 | 5% |

*3.6. Seed Germination*

The statistics showed that the seed germination of L-morph and S-morph was $13.33 \pm 3.87\%$ and $16.67 \pm 3.87\%$, respectively, with no significant differences between the seed germination of L-morph and S-morph ($p < 0.01$). We found widespread asexual clonal reproduction phenomena in the fields.

**4. Discussion**

This study revealed that *P. criopolitanum* has all the polymorphic intrinsic features with respect to style and stamen heights and accessorial characteristics with respect to pollen size and number. The polymorphic density of stigma papillae has recently been reported for the tristylous *Lythrumsalicaria* [25] and distylous *Polygonum jucundum* [12]. However, in this study, we found no significant differences between the L-morph and S-morph of

*P. criopolitanum*, including tepal size; thus, *P. criopolitanum* is typically distylous in intrinsic features and not in all ancillary features.

Flower morph frequencies have always received much attention [26], especially with respect to the consequences for inbreeding [27] or long-term population persistence [28]. In addition, discrepancies in the morph ratio have been found to be much higher in small rather than large populations of *Primula veris* [29] and *Primula elatior* [30], but populations solely composed of the L-morph or S-morph were seldom reported. Interestingly, we observed that the monotypic populations of *P. criopolitanum* were always found in wild populations, which perfectly accounted for the particularly low seed sets of the species.

Fluorescent microscopy showed that pin × pin crosses do occur at a low rate, and no pollen germination was identified in the case of the thrum self, which can explain slight deviations from the 1:1 ratio in a large population. Very common clonal growth in *P. criopolitanum* can explain why the populations fixed for the L-morph or S-morph were always distributed in the fields.

In distylous species, different taxa havebreeding systems with different compatibility levels. Most heterostylous species have a mating system with strict self-incompatibility; for example, a diallelic self-incompatibility system was reported in *Tylosema esculentum* through in vivo and in vitro diallel crossing experiments. The major site of pollen tube inhibition in the intramorph crosses was found to be in style [31]. *Arnebia szechenyi* has also been recorded to show heteromorphic self-incompatibility, which was further supported by the fact that no fruit was produced by flowers subjected to self-pollination or intramorph pollination [32]. In contrast, some species do not exhibit a strict self-compatibility system. In many species of *Primula*, selfing or crossing with a plant of the same morph will also produce a small numberof seeds [33]; for example, the fertility of a legitimate crossof *Primula merrilliana* was high, whereas the fertility of an illegitimate cross waslow [34], and *Pulmonaria officinalis* and *Ceratostigma willmottianum* were found to bepartially self-compatible [35,36]. Unlike most heterostylous species, *Primula oreodoxa* was found to be fully self-compatible under controlled self- and cross-pollinations [37]; flowers of *Psychotria carthagenensis* were also self-compatible [38], and atypical distylous *Psychotria goyazensis* was proven to be an intramorph self-compatible species [39]. In Europe, *Armeria maritima* is completely self-incompatible, but it lost its self-incompatibility during its migration to the New World through the Arctic regions [40]. Seed production is thought to be more sensitive to habitat fragmentation in heterostylous plants than in plants with other breeding systems because potential mating partners are more limited [41].

In this study, we nearly found no seed sets in most wild populations and very low seed sets occasionally occurred in large populations, which was different from other species of Polygonaceae; for example, the seed set of *Polygonum perfoliatum* has a high seed set rate (even up to 84%) [42], and bagging experiments have shown that 47% flowers of *Polygonum thunbergii* are self-pollinated because of no pollinator visits. Despite a high probability of cross-pollination, the probability of fruit set within the ramet was 0.30 because of resource limitations [43]. What interested us the most was that we found a similar scenario in *Polygonum viviparum*. The fruit set of *P. viviparum* has never been observed in North American populations, and sexual reproduction is clearly a rare event in this species [44]. The lack of viable seed production in *P. viviparum* has no single developmental explanation. A similar adaptive reproduction mechanism may exist in these species. On the basis of a common monotypic distribution, pollen germination during the stigma experiments, and the absence of seed sets in the wild populations, we inferred that *P. criopolitanum* has a strict self-incompatibility system.

Previous surveys of the incompatibility status of island flora such as the flora of New Zealand, Hawaii, and the Galápagos have shown a deficit in taxa with heteromorphic incompatibility when compared with continental areas [45]. We found that the dependence on water and environmental characteristics of the hydro-fluctuation belt may be important factors for the establishment of thereproduction system of *P. criopolitanum*. On the basis of the seed sets (natural and artificial conditions) and shoreside distribution along the

Yangzi River or other lakes, the strict self-incompatibility, as well as the sex reproduction efficiency, of *P. criopolitanum* was less dominant than asexual clonal reproduction and uniparental reproduction occurred by clone and not by self-compatible sex reproduction. Baker (1955) referred mainly to self-fertilization as a trait that would confer reproductive assurance during colonization, but we found a different scenario in *P.criopolitanum* [40]. Somes pecies have been reported to abandon sexual reproduction for some form of clonal reproduction, at least in some habitats or parts of their geographic range [46,47].

In conclusion, *P. criopolitanum* was typical distylous species with a strict self-incompatibility reproductive system. The wild population of *P. criopolitanum* deviated obviously from 1:1, and 1:1 style-morph ratios were occasionally found in very large populations. The strict self-incompatibility reproductive system, main environmental factors stress, and common monomorphic populations resulted in the low seed sets, which can explain the general asexual clonal reproduction instead of sexual reproduction in the species. To understand the molecular adaptation mechanism of *P. criopolitanum*, further studies on pollen flow and gene flow at the molecular level need to be performed.

**Author Contributions:** Project administration and writing, M.-L.C.; investigation, M.-Y.Q.; data curation, B.-B.B. and X.H. All authors have read and agreed to the published version of the manuscript.

**Funding:** The Natural Science Foundation of Anhui Province (1808085MC76) and the opening fund for provincial key laboratory and key discipline of Colleges of Life Sciences in AHNU, China.

**Institutional Review Board Statement:** Not applicable.

**Informed Consent Statement:** Not applicable.

**Data Availability Statement:** Population data and environmental and ecological factors data were gotten in 2021. Morphological data were gotten from February 2019 to September 2021. The data from the study is available upon request from authors.

**Conflicts of Interest:** The authors declare no conflict of interest.

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
