# Peer review of "Reproductive Ecology of Distylous Shoreside Polygonum criopolitanum Hance"

_diversity, doi:10.3390/d14030222_

Round 1
Reviewer 1 Report
The manuscript needs a certain revision before it can be considered for publication. My comments are below.
The title and abstract look to be well written. However, the list of keywords should be revised by avoiding words and sentences represented in the title.
Is there an acronym for the herbarium of Anhui Normal University? Please, add it to this herbarium collection.
On the map (Fig.1), please, add English "km" at the scale instead of (Russian?) "км".
In the section "Seed set experiments" (instead of "Seed sets experiments"), please, describe in detail the used modes used for the analysis.
Fig.5 looks to be superfluous. It is not necessary. Table 1 and Fig.4 describe properly the same results. As an alternative, you may move Fig. 5 to the Electronic Materials. The same is for Fig.6. These results are also included in Table 1.
In general, I strongly do not recommend using paragraphs represented by single sentences. This was found in Tables 2 and 3, and Fig.7. I suggest adding more information (description) of these tables and a figure.
CCA ordination was used incorrectly in such a case. CCA is usually used to find differences between analysed sites. In Fig.7, we don't see any designations of the dots or triangles, as well as it is not seen what differences are found between these symbols. If the main aim was to study correlations between environmental factors, I strongly suggest to use the correlation matrix instead of the CCA plot.
In lines 224-226, "The statistics showed that the pollen germination rate of pin style x thrum pollen was 72.0±4.0% (n=10) (Fig. 7-A)". But in Fig.7, there are no any A and B parts. However, if you mean Fig.8, this is already a mistake. In addition, Fig.8 doesn't present any statistics. This demonstrates only photos, not anything more. This relates to other parts with the wrong citations of "Fig. 7".
Fig. 9 looks to be superfluous. I suggest to delete it. In addition, the results of the section "3.6. seed germination" should be presented in another way. I suggest to present it as graph plot or as a table. In any case, to the results, please, add some statistical tests in these graphical elements (table or figure).
The text in lines 251-257 is recommended to be moved to the section Introduction. It is not related to the discussion or comparison of the results obtained in the study. Please, check this point through the whole text by leaving here only text, which is related to the discussion of the obtained results.
However, the remaining part of the Discussion is relatively well written with the comparison with the know references.
Finally, the section Conclusion is needed. Please, add this section to the manuscript.
Author Response
Please see the attachment below.

Reviewer 2 Report
The manuscript describing distyly in a species of Polygonum was interesting to read, and the authors present useful data demonstrating the presence of distyly in the species, which, as they show, can easily be overlooked. Much of the paper is descriptive, and this is useful in appropriately categorizing the breeding system in the genus and family. While I found the manuscript intriguing, there were a few areas that I thought could improve the manuscript.
1) The presence of many monomorphic populations brings up a number of questions concerning patterns of gene flow and reproduction, and the authors do not address this in much detail. This feels like a missed opportunity.
2) A number of the environmental characteristics are quite similar, if not the same, among the populations. It would be useful to see if the results change without the inclusion of these fairly static features. Additionally, the authors seem to be including qualitative data (i.e., habitat) along with quantitative data. Can the analytical method appropriately work with that type of data? I was under the impression that CCA could not, but I may be mistaken. I also do not feel as though the authors discussed the implication of these results.
3) I would use the word emasculation instead of castration.
4) How does the rate of seed germination compare to other species of Polygonaceae?
5) I don't quite follow the inclusion of Baker's Rule in the discussion. I would expect that according tp Baker's Rule, distylous species with heteromorphic SI would be less likely to colonize new areas compared to those that lack SI. Are the authors stating the asexually reproducing species are more likely to colonize new areas compared to those that are more reliant on sexual reproduction? How does the pattern of population distribution relate to the river system? Do the fruits/seeds water dispersed? Also, the geographic areas studied by the authors are all pretty close together, so is invoking Baker's Rule useful in the discussion (see https://nph.onlinelibrary.wiley.com/doi/10.1111/nph.13539)?
6) I think the English grammar needs to be improved.
7) If the authors are interested, there is other literature on SI (or lack thereof) on distylous species. It could provide a richer discussion to include more of that literature, particularly in the context of monomorphic populations.
Author Response
Please see the attachment below.

Round 2
Reviewer 1 Report
Dear Authors,
Thank you for considering my suggestions during the revision of the manuscript. I have read the responses provided by you in the letter. Based on them, I was satisfied with the corrections made during the revision of the manuscript.
I think that now the manuscript is ready for publication.
Author Response
I thanks a lot to the reviewer for giving me a lot of good advice. And I learned a lot of English!
Best wishes to the reviewer.
Chen Ming-Lin in Anhui Normal University
2022/3/3
Reviewer 2 Report
The manuscript is improved. There are a few additional changes or changes that weren't adequately addressed in this revision:
2.2.1 - number of pollen grains produced
3.2 - What is the " R-project test"?
3.3.3 - A qualitative features is included among the quantitative ones. Is this justified?
Author Response
Comments and Suggestions for Authors
The manuscript is improved. There are a few additional changes or changes that weren't adequately addressed in this revision:
2.2.1 - number of pollen grains produced
Response: To count the number of pollen grains produced. It means that we count the pollen number of a flower of L- or –S-morph. I have revised the sentence into “To count the pollen number per flower of L- or S-morph” in the manuscript.
3.2 - What is the " R-project test"?
Response: R-project test : To judge whether L- and S-morphs occurred at equal frequencies, we used R software (R Core Team. R: A language and environment for statistical computing, Vienna, 2019. https://www.R-Project.org.), I have revised the content in the manuscript.
3.3.3 - A qualitative features is included among the quantitative ones. Is this justified?
Response: I am sorry that I haven’t found the sentence “A qualitative features is included among the quantitative ones” in my manuscript. If tell me, I will revise it as soon as possible.